# An Integrative Analysis of the Transcriptome and Proteome of Rice Grain Chalkiness Formation Under High Temperature

**DOI:** 10.3390/plants13233309

**Published:** 2024-11-26

**Authors:** Shaolu Zhao, Ruijie Cao, Linhe Sun, Dongying Zhuang, Min Zhong, Fengli Zhao, Guiai Jiao, Pengfei Chen, Xinwei Li, Yingqing Duan, Xiaoxue Li, Shaoqing Tang, Shen Ni, Peisong Hu, Xiangjin Wei

**Affiliations:** 1State Key Laboratory of Rice Biology, China National Center for Rice Improvement, China National Rice Research Institute, Hangzhou 310006, China; zhaoshaolu313@163.com (S.Z.); caoruijie@caas.cn (R.C.); zhongmin2007@163.com (M.Z.); zhaofengli@caas.cn (F.Z.); jiaoguiai@caas.cn (G.J.); 82101215140@caas.cn (P.C.); lixinwei162013@163.com (X.L.); 18355093180@163.com (Y.D.); xx5990302@126.com (X.L.); tangshaoqing@caas.cn (S.T.); 2Jiangsu Coastal Area Institute of Agricultural Sciences, Yancheng 224002, China; 3Institute of Botany, Jiangsu Province and Chinese Academy of Sciences, Nanjing 210014, China; linhesun@cnbg.net; 4Xinyang Agricultural Experiment Station of Yancheng City, Jiangsu Academy of Agricultural Sciences, Yancheng 224049, China; zhuangdongying313@163.com; 5National Nanfan Research Institute (Sanya), Chinese Academy of Agricultural Sciences, Sanya 572025, China

**Keywords:** rice quality, high temperature, starch biosynthesis, transcriptome, proteome

## Abstract

Exposure to high temperatures can impair the grain-filling process in rice (*Oryza sativa* L.), potentially leading to the formation of chalky endosperm, but the molecular regulation mechanism remains largely elusive. Here, we reported that high-temperature (HT) stress (day/night, 35 °C/30 °C) reduces both the grain-filling rate and grain weight of Ningjing 1 variety compared to normal temperatures (NT, day/night, 28 °C/23 °C). Grains under HT stress exhibited an opaque, milky-white appearance, alongside significant alterations in starch physicochemical properties. An integrated transcriptomic analysis of grains under HT revealed up-regulation of genes related to defense mechanisms and oxidoreductase activity, while genes involved in sucrose and starch synthesis were down-regulated, and α-amylase genes were up-regulated. Proteomic analysis of grains under HT echoed this pattern. These results demonstrate that high temperature during the grain-filling stage significantly increases rice chalkiness by down-regulating genes related to sucrose and starch synthesis, while up-regulating those involved in starch degradation.

## 1. Introduction

Rice (*Oryza sativa* L.) is a staple food crop that feeds over half of the global population [1]. To meet the rising demand driven by population growth and economic development, global rice production must be increased by approximately 1% annually [2]. Unfortunately, high-temperature (HT) stress exacerbated by global climate change has emerged as one of the most detrimental environmental challenges to rice production. Each 1 °C rise in minimum temperature during the rice growing season results in a yield reduction of approximately 10% [3,4]. In addition to lowering yields, HT stress induces a higher grain chalkiness rate [5], a trait that not only affects rice appearance but also diminishes its milling and cooking properties, ultimately reducing its commercial value [6,7]. Therefore, understanding the mechanisms by which high temperature affects the rice endosperm filling process is critical for breeding heat-tolerant rice varieties and preserving rice yield and quality under HT stress.

Starch is the primary storage material in the rice endosperm, constituting over 80% of its total dry weight. Structurally, starch in the rice endosperm is categorized into amylose, which consists of linear α-1,4-glycosidic bonds, and amylopectin, a highly branched molecule with α-1,6-glycosidic bonds. Starch synthesis in the rice endosperm is regulated by several key enzymes, including ADP-glucose pyrophosphorylase (AGP), granule-bound starch synthase (GBSS), soluble starch synthase (SSS), starch branching enzyme (SBE), and starch debranching enzyme (DBE) [8,9,10]. Chalkiness refers to the opaque portions of the rice endosperm, and HT during grain filling is a key environmental factor promoting chalkiness formation [11]. Previous research has shown that the increase in chalky grains under HT stress is linked to altered expression of starch-synthesis genes or enzymatic activity [12,13]. During grain filling, HT stress decreases the expression of *GBSSI* and *SBEIIb*, inhibiting enzyme activities and leading to a lower amylose content (AC) and higher gelatinization resistance [14]. At the ripening stage, the isoforms of SBE genes (*SBEI*, *SBEIIb*, and *SBEIII*) are down-regulated under HT, while isoforms of SSS genes (*SSSIIb*, *SSSIIc*, *SSSIIIb*, and *SSSIVa*) are up-regulated [14]. *SSIIIa*-RNAi significantly impacts chalkiness under HT but has little effect at normal temperatures [15]. Moreover, HT induces atypical starch degradation rather than starch synthesis. Suppressing α-amylase genes can reduce the amylase activity and decrease chalky grain formation under HT [13,16].

High temperatures during the grain-filling period cause starch granules to become loosely packed, decreasing the starch content in dispersed granules and resulting in reduced grain weight and increased chalkiness [17]. Previous studies have shown that for japonica rice varieties, temperatures above 26 °C during grain filling can induce chalky grains and decrease yield [13,18]. Chalkiness is a complex trait regulated by quantitative trait loci (QTLs). Over 100 QTLs associated with chalky grains have been mapped across the 12 rice chromosomes [19], with several identified as temperature-sensitive. Three QTLs for chalkiness in brown rice under HT stress were detected using two japonica rice varieties [20], and four QTLs (*qWK1-1*, *qWK1-2*, *qWK2*, and *qWK8*) were identified in a recombinant inbred line (RIL) population exposed to HT [21]. Two QTLs, *qPGC9* and *qPGC11*, were fine-mapped to chromosomes 9 and 11, respectively, and were found to be sensitive to HT. Additionally, the QTL *Apq1*, which regulates chalkiness under HT tolerance, was fine-mapped to a 19.4 kb region. The thermo-responsive allele of *SUCROSE SYNTHASE 3* (*SUSY3*) is likely the causal gene for *Apq1* [22,23]. The QTL *OsSFq3*, associated with HT tolerance and chalkiness, was fine-mapped to a 7.1 cM interval on chromosome 3, and the causal gene for *OsSFq3* was identified as *LOC_Os03g48170* [24]. Despite these advances, the molecular mechanisms underlying the formation of chalky endosperm in rice under HT remain poorly understood.

Here, we performed HT (day/night, 35 °C/30 °C) and NT (day/night, 28 °C/23 °C) treatments of the japonica rice cultivar Ningjing 1 at the booting stage in controlled growth chambers. HT stress resulted in a decreased grain-filling rate and grain weight, an opaque, milky-white grain appearance, and significant changes in starch physicochemical properties. Transcriptomic and proteomic analyses revealed down-regulation of genes involved in sucrose and starch synthesis and metabolism, along with up-regulation of genes related to starch degradation, defense, and oxidoreductase activity. These findings provide valuable insights into the mechanisms of HT stress affecting rice quality and yield, laying a foundation for breeding rice varieties with improved heat tolerance.

## 2. Results

### 2.1. High-Temperature Stress Lead to Grain Incomplete Rate and Increased Floury Grains in NJ1

The grain-filling rate of cv. NJ1 plants subjected to high-temperature (HT) stress was significantly slower than that of plants grown under normal-temperature (NT) conditions from 10 days after fertilization (DAF; Figure 1A,B). This slower filling rate resulted in a marked reduction in grain width and thickness (Figure 1C), and grain length was not significantly affected by high-temperature stress (Figure 1C). In contrast to the transparent grains under NT, the mature endosperm of HT-exposed plants appeared opaque and milky-white (Figure 1D,E). Scanning electron microscopy (SEM) of transverse endosperm sections showed densely packed, uniformly sized polyhedral starch granules (SGs) in HT plants, while NT plants exhibited loosely packed, rounder, smaller, and irregularly arranged compound SGs (Figure 1F). Overall, HT adversely impacted grain filling and led to the typical symptoms of chalky endosperm.

### 2.2. High Temperature Impacts the Physicochemical Properties of Rice Starch

Given the significant effects of HT on both grain-filling rate and grain appearance, we analyzed the physicochemical properties of starch in mature grains under both NT and HT conditions. Consistent with the lower filling rate, HT significantly reduced total starch, amylose, and amylopectin per grain compared to NT (Figure 2A–C). To further assess changes in amylopectin fine structure, we debranched endosperm starch with isoamylase and analyzed the chain length distribution. Under HT, the proportion of chains with a degree of polymerization (DP) of 6 to 16 significantly decreased, while chains with a DP of 17 to 60 obviously increased (Figure 2D).

We also examined the pasting properties of the starch using a rapid visco analyzer (RVA). The RVA profile revealed that HT-treated starch exhibited a trend similar to NT-treated starch, though at consistently lower levels. Peak viscosity and final viscosity values were significantly lower under HT compared to those under NT (Figure 2E). Gelatinization temperature of endosperm starch was also analyzed by a differential scanning calorimeter (DSC), and the result showed that the gelatinization enthalpy (ΔH) of starch under HT were significantly higher than that under NT (Figure 2F). Starch solubility in urea solutions was also tested, revealing that HT-treated starch was more resistant to gelatinization in 6 M to 9 M urea, while starch gelatinization was easier under NT (Figure 2G). In summary, HT markedly altered the physicochemical characteristics of starch compared to at NT.

### 2.3. Genes and Proteins Involved in Sucrose and Starch Synthetic Metabolism Were Regulated by HT Stress

To understand the genetic and protein-level responses of rice to HT stress during the grain-filling stage, we conducted integrated transcriptomic and proteomic analyses. A total of 950 genes and 1183 proteins were significantly affected by HT (Figure 3A; Appendix A). Among these, 26 genes were up-regulated, and 54 genes were down-regulated at both transcript and protein levels under HT (Appendix A). Interestingly, more genes were altered at the protein level than at the transcript level, with 80 genes showing changes at both levels. Notably, more genes were down-regulated at the protein level, while more were up-regulated at the transcript level, likely due to the delayed nature of translation (Figure 3A,B).

Pathway annotation of differentially expressed genes (DEGs) and proteins (DEPs) revealed that the “starch and sucrose metabolism” and “nitrogen metabolism” pathways were significantly enriched under HT stress, indicating a strong impact of HT on starch synthesis (Figure 3C). Specifically, 46 of 437 genes and 46 of 102 proteins involved in starch and sucrose metabolism were significantly differentially expressed under HT (Figure 3C). This suggests that these genes play a pivotal role in the HT response, with key starch-synthesis genes down-regulated under HT stress potentially contributing to the reduced grain-filling rate.

Gene Ontology (GO) and KEGG pathway analyses indicated that HT-responsive genes were primarily involved in metabolic pathways related to carbohydrate transport and metabolism, protein transport and metabolism, and defense (Figure 4, Appendix A). Carbohydrate and protein transport/metabolism-related genes were mostly down-regulated, while genes involved in energy production and conversion were positively correlated (Figure 4). Conversely, defense-related genes were induced under HT stress.

### 2.4. Starch or Sucrose Synthesis Was Repressed and the Hydrolyzation Was Enhanced Under High-Temperature Stress

Based on integrated transcriptome and proteome analyses, genes related to starch- or sucrose-synthesis pathways were primarily significantly down-regulated under high-temperature stress in both transcriptome level and proteome level, such as *SuSy2*, *SuSy3*, *OsAGPL1*, *SSII-1,3*, *OsAGPS2b*, and *Wx* were significantly down-regulated at both transcript and protein levels under HT. Starch-degradation genes, particularly *α-Amy3E*, were notably up-regulated (Figure 5, Appendix A). The related transport proteins also exhibit the same expression pattern, such as BT1 and GPT1. However, PGI, which catalyzes Fru-6P into Glc-6P, was not significantly regulated in both transcriptome level and proteome level. Interestingly, starch-synthesis-related genes like *SuSy1*, *SuSy4*, and *OsAGPL2* were suppressed only in proteome level and not transcriptome level. Conversely, key genes in the starch-degradation pathway were significantly up-regulated under high temperature stress, such as *α-Amy3E* (Figure 5, Appendix A). Taken together, the enhancing of the starch-degradation pathway and suppression of the synthesis pathway could be the main reason for low starch content in rice seeds and the increasing chalkiness in grains under the high-temperature stress.

### 2.5. High Temperature Disturbs the Expression and Translation of Genes Related to Starch Synthesis

Western blot and qRT-PCR analyses confirmed the transcriptomic and proteomic findings. The protein abundance of key starch-synthesis enzymes such as GBSSI, OsAGPL1, OsAGPS2b, OsAGPS1 in NJ1 developing endosperm were significantly decreased under HT stress, with a noticeable reduction in SSIIa (Figure 6). qRT-PCR results also showed that the expression of *LEA3* and *BXL7* was significantly increased, while most other genes, including the key starch-synthesis enzymes *BT1*, *PRMS*, *LacZ*, *HEXO3*, *LEA3*, *BXL6*, *BXL7*, *KPYA*, *CDPK1*, *UVR8,* and *UBC2* were down-regulated. These findings are consistent with the integrated transcriptomic and proteomic analyses.

## 3. Discussion

High temperature (HT) during the ripening stage is widely recognized as a key factor leading to increase the chalky grain appearance, reduced grain weight and starch content of rice. Multiple studies have examined the morphological features of chalky grains and identified loosely packed, round starch granules that form air spaces, altering light refraction in chalky grains [25,26,27,28]. The density of these starch granules in chalky grains (or chalky portions) is notably lower than in translucent grains (or translucent portions), making chalky grains more prone to breakage during milling, which in turn reduces rice processing quality [29]. Occasionally, small pits have been observed on the surface of the round starch granules in HT-induced chalky grains [30]. Consistent with these observations in previous research, our study found that the chalky grains of NJ1 subjected to HT stress exhibited a similar structure, with loosely arranged, round starch granules and substantial air spaces between them (Figure 1F). Additionally, the grain-filling rate of NJ1 was significantly slower under HT than that under normal temperature. Both amylose and amylopectin content in the endosperm of grains were significantly reduced under HT compared to those under NT conditions.

The incomplete accumulation of starch is thought to be the primary cause of chalkiness. Many studies on HT-induced chalkiness in grains have focused on genes involved in starch synthesis or carbon metabolism. Rice endosperm primarily stores two types of starch: linear α-polyglucan amylose and branched α-polyglucan amylopectin. Amylose synthesis is mainly regulated by GBSS, while amylopectin synthesis is catalyzed by several isoforms of enzymes, including soluble starch synthase, starch branching enzyme, and starch debranching enzyme [31]. Starch-synthesis genes can be categorized into two types based on their location: type I is active in seeds and sink tissues, while type II functions in vegetative tissues and source organs. *GBSSI*, *SSI*, *SSIIa*, *SSIIIa*, *BEI*, *BEIIb*, *ISA1*, *ISA3*, and *PUL* are preferentially and specifically expressed in rice endosperm [32,33]. The functions of SSI, SSIIa, SSIIIa, BEI, BEIIa, and BEIIb in synthesizing different amylopectin chains under HT have been elucidated through physicochemical analyses of mutants. For example, double mutants of SSIIa (alk) and BEIIb (amylose-extender, ae) significantly increased amylose content, altered amylopectin chain length distribution, and lowered the gelatinization temperature of starch granules [34,35]. Previous research has shown that HT during the grain-filling stage reduces amylose content and increases the proportion of long amylopectin chains in the rice endosperm [25,36,37,38]. Transcription of *GBSSI* and *BEs* were significantly inhibited during grain filling under HT, while genes for α-amylase and starch-degrading enzyme were transiently induced, which may be the reason of the pits and round surfaces in the starch granules of HT-ripened grains [5,16] (Figure 6). Moreover, our transcriptome and proteomic analysis of grains under HT echoed this pattern.

In this study, we found that the amylose content in the developing endosperm of NJ1 was significantly decreased under HT stress. Additionally, short amylopectin chains (DP < 16) were reduced, while middle and long chains (DP > 17) were increased. The gelatinization temperature was also lower under HT compared to that under NT. Comprehensive transcriptomic and proteomic analyses revealed that genes involved in starch or sucrose metabolism pathways were largely down-regulated under HT stress. Western blot and qRT-PCR analysis further confirmed that genes associated with starch synthesis were down-regulated at both the transcriptional and translational levels under HT. Comprehensive transcriptomic and proteomic analyses indicated that HT stress not only altered the expression of genes related to starch and sucrose metabolism in the developing endosperm of NJ1, particularly by down-regulating starch metabolism genes, but also reduced the expression of ADP-Glc and Glc-6-P transporters. These disruptions in starch accumulation and physicochemical properties in the endosperm led to the formation of chalky grains. Heat shock proteins (HSPs) play a vital role in plant growth and development under stress conditions. Previous studies observed high transcription levels of OsHSP17.9A and OsHSP26.7 during seedling and anthesis stages. In our study, 12 heat-shock proteins were significantly differentially expressed under HT stress, with 10 of them being notably higher expressed under HT treatment (Appendix A). Consequently, our research provides valuable heat-tolerance gene resources and strategies for breeding rice varieties with improved heat tolerance.

## 4. Conclusions

In summary, HT stress resulted in a decreased grain-filling rate and grain weight; an opaque, milky-white grain appearance; and significant changes in starch physicochemical properties. Transcriptomic and proteomic analyses revealed down-regulation of genes involved in sucrose and starch synthesis and metabolism, along with up-regulation of genes related to starch degradation, defense, and oxidoreductase activity. These findings provide valuable insights into the mechanisms of HT stress affecting rice quality and yield and provide a valuable gene resource and strategy for genetic improvement of rice grain quality.

## 5. Materials and Methods

### 5.1. Plant Materials and Growth Conditions

The japonica rice cultivar NJ1 was used for all experiments. During the normal growing season, the plants were cultivated in a paddy field at the China National Rice Research Institute (119°55′ E, 30°04′ N). At the booting stage, the plants were transferred to growth chambers with a 12 h photoperiod and 75% humidity. Two temperature treatments were applied: high-temperature treatment (HT) with a day–night temperature of 35 °C/30 °C, and normal-temperature treatment (NT) with a day–night temperature of 28 °C/23 °C. Grains at 10 DAF were snap-frozen in liquid nitrogen and stored at −80 °C for transcriptomic and proteomic analyses. Mature grains were used for biochemical and electron microscopy studies. Whole grains were selected to measure grain size (grain length, width, and thickness) and 1000-grain weight. Grains at various developmental stages (3–30 DAF) in both HT and NT treatments were harvested for phenotypic observations and dry weight measurements (100 grains, repeated three times).

### 5.2. Scanning Electron Microscopy (SEM) Analysis

The mature grains harvested from HT and NT plants were transversely cut using a knife, and the cross-sectional surface of the fracture was coated with gold. Images were captured using a HITACHI S3400N scanning electron microscope (https://www.hitachi-hightech.com/). The SEM protocol followed the method described in a previous study. The plant materials used for SEM were not less than three biological replicates [39].

### 5.3. Determination of Physicochemical Properties of Starch

Flour from whole grains harvested from HT and NT plants was used to measure total starch, amylose, and protein content. The total starch was measured using a starch assay kit (Megazyme, Wicklow, Ireland), following the manufacturer’s protocol. The amylose content was determined using the commercial enzymatic kit K-AMYL (Megazyme), as described by Wei et al. [40]. The protein content was measured using the method described by Kang et al. [39]. To assess the effect of high temperature on amylopectin chain length distribution, 5 mg of rice flour from both HT and NT plants was digested with isoamylase (Megazyme) and analyzed using capillary electrophoresis (PA800 plus pharmaceutical analysis system, Beckman Coulter, Brea, CA, USA). The pasting properties of flour milled from HT and NT grains were analyzed using a Rapid Visco Analyzer (RVA Techmaster; Newport Scientific, Sydney, Australia), following the manufacturer’s instructions. The swelling and gelatinization properties of endosperm starch in urea solution were determined using the method described by Nishi et al. [36]. All tests were performed in three biological replicates.

### 5.4. RNA Extraction and qRT-PCR Analysis

Total RNA was extracted from rice endosperm at 10 DAF from HT and NT plants using Trizol reagent (Thermo Fisher Scientific, San Jose, CA, USA), following the manufacturer’s instructions. The RNA treated with DNase I was reverse-transcribed into cDNA using the ReverTra Ace qPCR-RT Kit (Toyobo, Osaka, Japan) with oligo(dT) primers. Quantitative real-time PCR (qRT-PCR) was performed with SYBR Green Real-time PCR Master Mix (Toyobo, Osaka, Japan) using the 2^−∆∆CT^ method. The rice *Ubiquitin* gene (*Os03g0234200*) was used as an internal control. The primers used for qRT-PCR are listed in Appendix A.

### 5.5. RNA-Seq, Data Processing, and Gene Annotation

The extracted RNA in 4.4 was treated with DNase I (Takara, Kyoto, Japan) for 30 min at 37 °C to remove genomic DNA. RNA quality was verified using a Bioanalyzer 2100 and an RNA 6000 Nano LabChip kit (Agilent, Santa Clara, CA, USA), with a minimum RNA Integrity Number (RIN) value > 7.0, as described by Zhou et al. [41]. Approximately 10 µg of total RNA was used to remove ribosomal RNA fragments with a Ribo-Zero^®^ rRNA Removal Kit (Plant Seed/Root; Illumina, San Diego, CA, USA), as described by Yue et al. [42]. The purified RNA was fragmented by adding divalent cations at elevated temperatures. The fragmented RNA was reverse-transcribed to create cDNA sequencing libraries using an Illumina mRNA-Seq sample preparation kit. Paired-end sequencing of samples from HT and NT plants, each with three biological replicates, was conducted by LC-Bio (Hangzhou, China) using an Illumina Hiseq 4000 system. Gene annotation was performed using the Blast2GO program, and GO enrichment analysis identified significantly enriched biological functions among differentially expressed genes (DEGs). Metabolic pathway assignments were made using the Kyoto Encyclopedia of Genes and Genomes (KEGG) annotation.

### 5.6. Proteomic Analysis

The proteins were extracted from rice endosperm at 10 DAF from HT and NT plants. A total of 0.2 g of starchy endosperm was ground in liquid nitrogen and suspended in lysis buffer (7 M urea, 2 M thiourea, 4% CHAPS, 40 mM Tris-HCl, pH 8.5). After 5 min, 10 mM DTT was added. The samples were sonicated at 200 W for 15 min, centrifuged at 4 °C (30,000g for 15 min), mixed with 5 volumes of chilled acetone containing 10% TCA, and incubated at −20 °C overnight. The precipitates were washed with chilled acetone three times and resuspended in 8 M urea. Protein concentration was measured using the Pierce 660 nm Protein Assay Kit (Thermo Fisher Scientific, San Jose, CA, USA) with bovine serum albumin (BSA) as a standard. For digestion, 100 µg of protein was treated with Trypsin Gold (Promega, Beijing, China) at a protein ratio of 30:1 at 37 °C for 16 h. The peptides were dried, reconstituted, and labeled using an 8-plex iTRAQ reagent (Applied Biosystems, Foster City CA, USA). The samples were labeled as follows: NT1 (113), NT2 (114), NT3 (115), HT1 (117), HT2 (118), and HT3 (119). The labeled peptides were pooled and dried by vacuum centrifugation. The peptide mixtures were fractionated by SCX chromatography (Shimadzu LC-20AB HPLC system, Shimadzu, Kyoto, Japan) and collected every minute. Each fraction was desalted, vacuum-dried, and resuspended for further analysis using nanoHPLC (Shimadzu LC-20AD, Shimadzu, Kyoto, Japan) and a C18 column. Data from Orbitrap were converted into MGF files for protein identification using Mascot (Matrix Science, Mascot Distiller2.7), with searches against the NCBInr Oryza sativa and UniProtKB/Swiss-Prot databases. Protein quantification was performed using Mascot, with stringent criteria applied for protein identification and fold-change determination (>1.2, *p*-value < 0.05). The differentially expressed proteins were classified using GO and KEGG databases.

### 5.7. Protein Extraction and Western Blot Analysis

The total protein from rice endosperm at 10 DAF was extracted as described by Kang et al. [41]. The proteins were resolved by SDS-PAGE, transferred to PVDF membranes, and immunoblotted with specific antibodies. Detection was performed using the ECL Plus Western Blotting Detection Kit (Thermo Fisher Scientific, San Jose, CA, USA), and signals were visualized with a ChemiDoc MP system (Bio-Rad, Hercules, CA, USA).

### 5.8. Data Analysis

The data are presented as the means ± standard deviation (SD), shown by error bars. The *t*-test (* *p* < 0.05; ** *p* < 0.01; NS, not significant) were used for statistical analysis using Office 2016 software.

## Figures and Tables

**Figure 1 plants-13-03309-f001:**
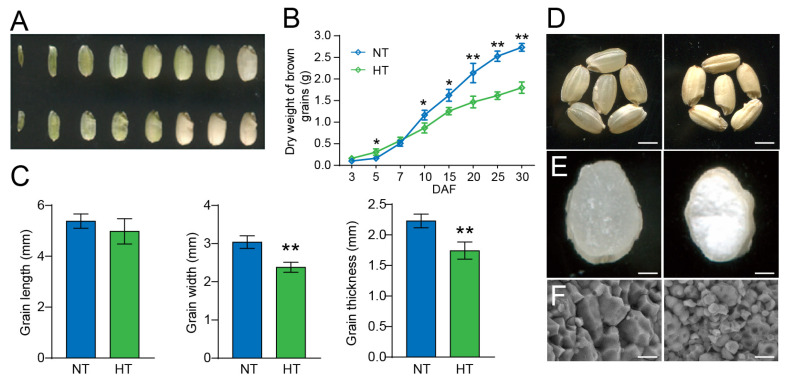
Characterization of cv. NJ1 under normal-temperature (NT) and high-temperature (HT) treatment. (**A**) Dry brown grains of NJ1 under NT (above) and HT (below) at various stages of development. (**B**) Weight of dry brown grains of NJ1 under NT and HT at various stages of grain filling. DAF, days after fertilization. (**C**) Grain length, width, and thickness of NJ1 under NT and HT. (**D**) Comparison of phenotype of cross-sections of NJ1 mature brown grains between NT and HT; scale bar, 5 mm. (**E**) Phenotypic comparison of the NJ1 brown grains between NT and HT; scale bar, 5 mm. (**F**) Scanning electron microscopy (SEM) images of transverse sections of NJ1 endosperm under NT and HT; scale bar, 10 µΜ. Data in (**B**,**C**) are presented as mean from three replicates. Asterisks indicate statistical significance between NT and HT determined by Student’s *t*-tests (* *p* < 0.05; ** *p* < 0.01).

**Figure 2 plants-13-03309-f002:**
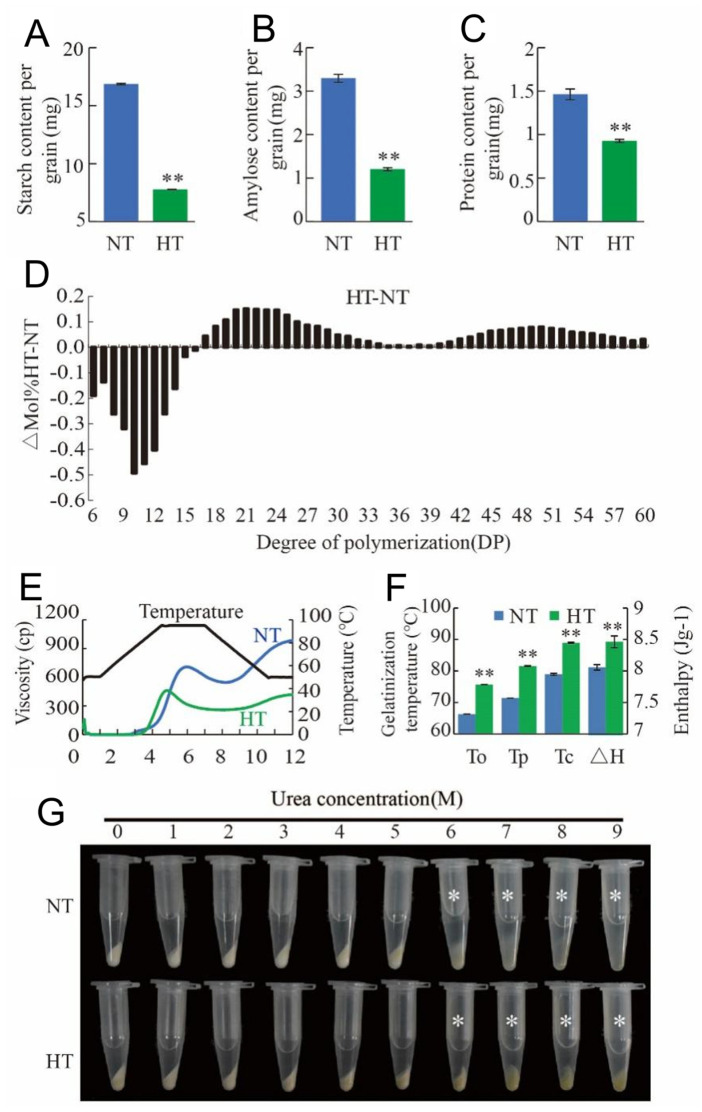
Physicochemical characteristics of NJ1 starch in NT and HT. (**A**–**C**) Contents of total starch (**A**), amylose and (**B**) protein (**C**) per grain in NT and HT. (**D**) Difference in amylopectin chain length distribution of NJ1 starch between NT and HT. (**E**) Pasting properties of NJ1 endosperm starch in NT and HT. Black line indicates temperature changes during measurement. (**F**) Gelatinization temperature of endosperm starch in NT and HT. TO, TP, TC, and △H represent onset, peak, conclusion gelatinization temperature, and enthalpy, respectively. (**G**) Urea dissolving properties of starch in NT and HT. Starch powder was mixed with different concentrations (1 to 9 M) of urea solution. Asterisks indicate starch in HT is more difficult to gelatinize in 6–9 M urea solution than that of NT. All data are presented as mean = ±SD from three replicates. Asterisks indicate statistical significance between NT and HT determined by Student’s *t*-tests (* *p* < 0.05; ** *p* < 0.01).

**Figure 3 plants-13-03309-f003:**
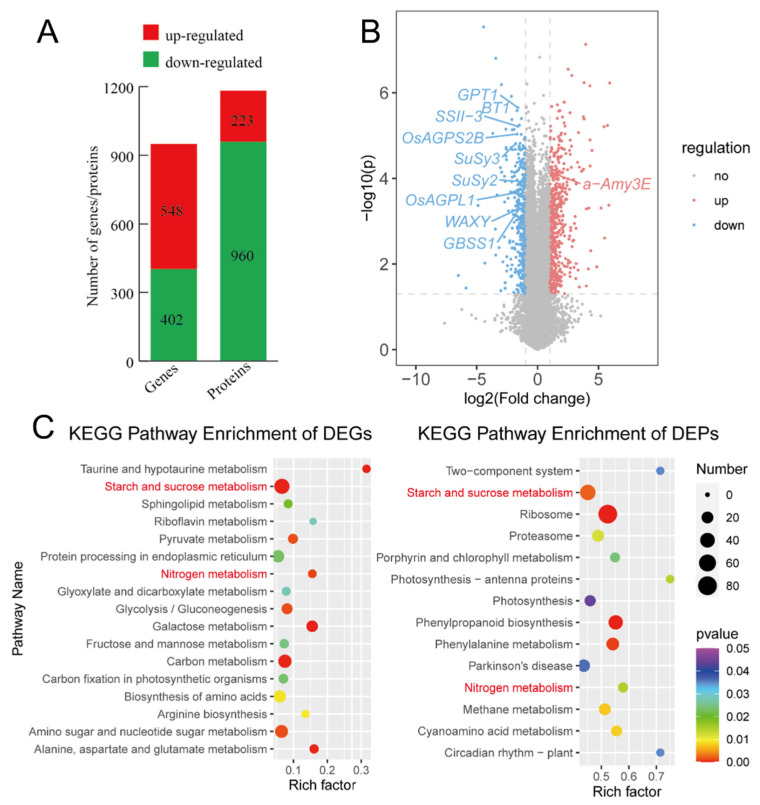
Differentially expressed genes and proteins in HT as compared to those in NT. (**A**) Numbers in columns are numbers of genes/proteins. (**B**) Volcano plot of transcripts. Red points indicating up-regulated genes, blue ones indicating down-regulated genes. Genes in starch-synthesis pathway are labeled. (**C**) KEGG enrichment of both different expression genes (left) and proteins (right) with *p* value < 0.05. Pathways enriched in both genes and proteins were marked as red.

**Figure 4 plants-13-03309-f004:**
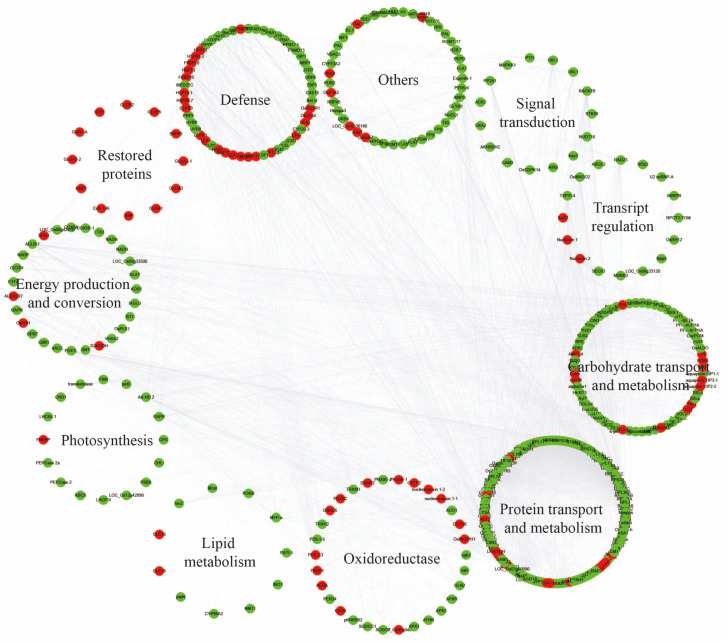
GO and KEGG analyses of gene and protein responses to high temperature. Differentially expressed genes/proteins are shown in letters with red and green backgrounds, respectively, to indicate either a rise or a fall in abundance. Full forms of the abbreviated ID’s are given in Appendix A.

**Figure 5 plants-13-03309-f005:**
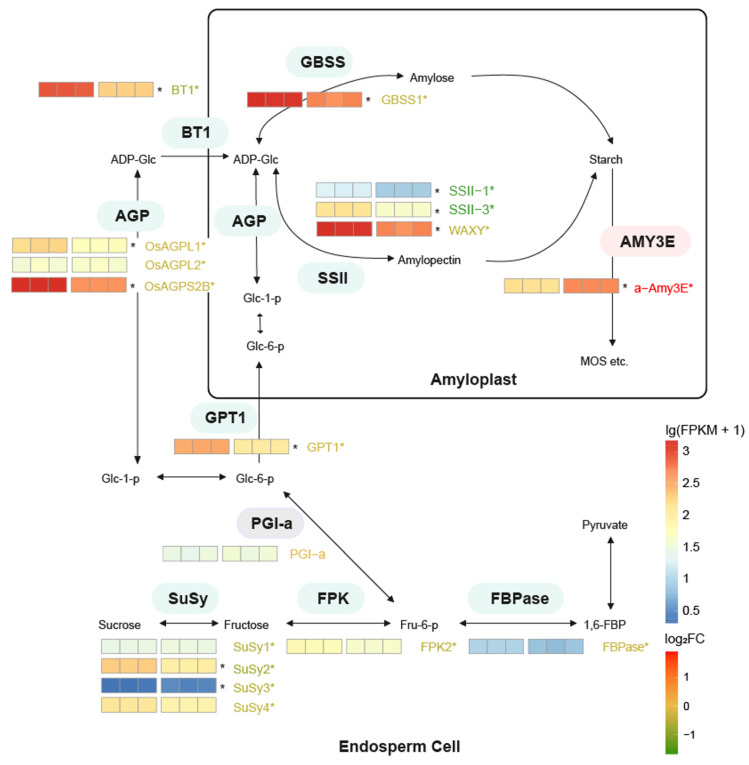
The differences of gene expression or protein abundance related to starch or sucrose metabolic pathways involved in response to high temperature. The red and green backgrounds indicate either a rise or a fall in abundance (grey indicates no significant differences). lg(FPKM + 1) of genes are shown as heatmaps. The asterisks on the right of heatmaps indicate the significant differences on genes expressed level (*p* < 0.05). The labels beside heatmaps indicate corresponding proteins, and its color indicates the log2(fold-change) of proteins, the asterisks beside the labels indicate significant differences of proteins abundance (*p* < 0.05).

**Figure 6 plants-13-03309-f006:**
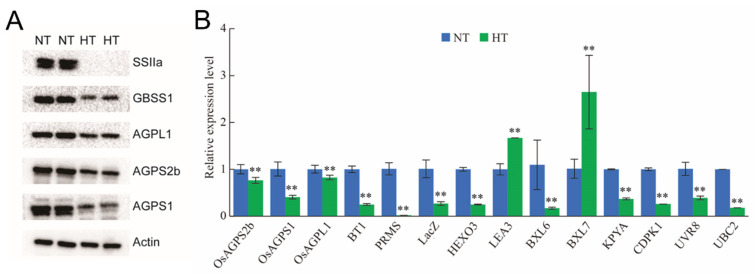
The confirmation of the differential abundance of genes and proteins obtained by, respectively, Western blot and qRT-PCR analysis. (**A**) The Western blot analysis of the differentially expressed proteins related to starch metabolism. (**B**) The qRT-PCR analysis of differential abundance of genes in HT as compared to those in NT. Values are means ± SD (n = 3); the asterisks indicate statistical significance between the HT and NT, as determined by Student’s *t*-test (** *p* ≤ 0.01). The rice genes Actin and Ubiquitin were used as the reference sequences.

## Data Availability

The datasets supporting the conclusions of this article are included within the article and Appendix A.

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
