# Peer review of "An Integrative Analysis of the Transcriptome and Proteome of Rice Grain Chalkiness Formation Under High Temperature"

_plants, 2024, doi:10.3390/plants13233309_

Round 1
Reviewer 1 Report
Comments and Suggestions for Authors
The research topic is interesting and is in line with current demands for plant material adapted to extreme conditions; its results are relevant within the genetic improvement program of the species.
Some minor corrections are noted in the manuscript that contribute to improving the understanding of the manuscript, and some are mentioned below:
1. The title is consistent with the nature of the study.
2. In the abstract, it is suggested to include the objective, the methodologies and end with the impact at the biological, genetic and agronomic level of the main findings of the research.
3. Keywords are appropriate for this type of study.
4. The introduction presents the fundamental elements of the research; however, it is necessary that the previous studies be contextualized and updated.
5. The methodologies applied are the most appropriate for this type of study, however, it is necessary to include the variables and statistical analyses and the software used for this.
6. The results are presented in a very descriptive manner within the manuscript, some tables are indicated that should be included within it, not as supplementary material.
7. The discussion addresses all the elements found in the results, however, it is suggested that it be complemented with contextualized comparative studies, at international or regional level, as well as mentioning the species. It is important that the discussion highlights the implications at the biological, genetic and agronomic level of the results found.
8. The conclusion is accurate and consistent with the scope of the research, however, more emphasis could be placed on the practical implications of the results found.
9. Authors are encouraged to use references from 2019 onwards, if possible.
10. Improve the resolution of some figures.

Author Response
The research topic is interesting and is in line with current demands for plant material adapted to extreme conditions; its results are relevant within the genetic improvement program of the species.
Some minor corrections are noted in the manuscript that contribute to improving the understanding of the manuscript, and some are mentioned below:
Response: Thanks very much for your comments and good suggestions.
- The title is consistent with the nature of the study.
Response: Thanks for your affirmation.
- In the abstract, it is suggested to include the objective, the methodologies and end with the impact at the biological, genetic and agronomic level of the main findings of the research.
Response: Thanks a lot for your constructive suggestion. We have re-arranged those part in the revised abstract.
- Keywords are appropriate for this type of study.
Response: Thanks for your affirmation.
- The introduction presents the fundamental elements of the research; however, it is necessary that the previous studies be contextualized and updated.
Response: Thank you for pointing this out. We have updated some reference to make the description more contextualized in the introduction part as you suggested.
- The methodologies applied are the most appropriate for this type of study, however, it is necessary to include the variables and statistical analyses and the software used for this.
Response: Thank you for pointing them out. We have included it as you suggested in Materials and methods part in line 378-381 (5.8 Data analysis).
- The results are presented in a very descriptive manner within the manuscript, some tables are indicated that should be included within it, not as supplementary material.
Response: Thank you for your kind suggestions. Due to the large number of primers, genes, proteins listed in the supplementary tables, it is difficult to list them all in the text. After consideration, we have reserved them as supplementary tables, it also easy to search and download for viewers.
- The discussion addresses all the elements found in the results, however, it is suggested that it be complemented with contextualized comparative studies, at international or regional level, as well as mentioning the species. It is important that the discussion highlights the implications at the biological, genetic and agronomic level of the results found.
Response: Thanks a lot for your constructive suggestion, we have revised the discussion in line 223-280.
- The conclusion is accurate and consistent with the scope of the research, however, more emphasis could be placed on the practical implications of the results found.
- Authors are encouraged to use references from 2019 onwards, if possible.
Response: Many thanks for your suggestion, we have checked and updated most of the references as you suggested.
- Improve the resolution of some figures.
Response: Thank you for pointing it out. We have improved the resolution of some figures in the revised manuscript.
Reviewer 2 Report
Comments and Suggestions for Authors
1. The information contained in lines 79-89 is located in the Introduction section; however, its content closely resembles a description of the methodology with some key results and implications (Results and discussion). Please relocate it to a more relevant place within the manuscript.
2. Results. Lines 96-97. It should be clearly stated that grain length was not a trait significantly affected by high-temperature stress, as there was no statistically significant difference between the two temperature regimes, as shown in Figure 1C. Numerical differences are not sufficient for such claims.
3. A section for statistical analysis of the information should be included in the Materials and Methods section.
4. It is suggested that the Materials and Methods section be placed immediately after the Introduction to enable the reader to follow a more logical reading flow.
5. Conclusions. Lines 280, 281 and part of 282 do not refer to conclusions but to materials and methods, and contain information provided previously. Suggested to delete.
Author Response
- The information contained in lines 79-89 is located in the Introduction section; however, its content closely resembles a description of the methodology with some key results and implications (Results and discussion). Please relocate it to a more relevant place within the manuscript.
Response: Many thanks for your suggestion, we have revised this content in the introduction section, thus it is more appropriate. Please check it in lines 81-87.
- Results. Lines 96-97. It should be clearly stated that grain length was not a trait significantly affected by high-temperature stress, as there was no statistically significant difference between the two temperature regimes, as shown in Figure 1C. Numerical differences are not sufficient for such claims.
Response: Thank you for pointing them out. We have corrected it as grain length was not significantly affected by high-temperature stress as you suggested in line 96-97.
- A section for statistical analysis of the information should be included in the Materials and Methods section.
Response: Thank you for your good suggestions. We. have added this part in 5.8 Data analysis in Materials and Methods section in line 378-381.
- It is suggested that the Materials and Methods section be placed immediately after the Introduction to enable the reader to follow a more logical reading flow.
Response: Thank you for pointing them out. We have corrected it as you suggested in revised manuscript.
- Conclusions. Lines 280, 281 and part of 282 do not refer to conclusions but to materials and methods, and contain information provided previously. Suggested to delete.
Response: Thanks for your suggestion. We have deleted it.
Reviewer 3 Report
Comments and Suggestions for Authors
The work presented here fits closely into a very important area related to the cultivation of food crops under changing environmental conditions. These changes seem to be inevitable in the coming decade in view of climate change on Earth.
The article prepared focuses on the analysis of the tanscriptome and proteome, which seems to be a very important research direction.
The team presented work carried out using modern research techniques, and the description of the procedures used allows a thorough analysis of the experimental work carried out and demonstrates good preparation for the planned experiments.
The results presented in the paper are for the most part unquestionable from the point of view of their presentation; the well-prepared graphs and drawings, as well as the illustration with photographs, allow the reasoning to be followed. Only the representation of the level of difficulty is gelatinize, by means of photographs seems to be relatively unreliable - perhaps any method determining the level of turbidity would have worked better.
I also very much appreciated the preparation and description of Figure 5 aimed at showing the relationships in sucrose metabolic pathways involved in response to high temperature.
The well-prepared discussion, referring to other literature data, shows a critical approach to the results obtained.
In my opinion, the article presented is worthy of publication.
Author Response
The work presented here fits closely into a very important area related to the cultivation of food crops under changing environmental conditions. These changes seem to be inevitable in the coming decade in view of climate change on Earth.
The article prepared focuses on the analysis of the transcriptome and proteome, which seems to be a very important research direction.
The team presented work carried out using modern research techniques, and the description of the procedures used allows a thorough analysis of the experimental work carried out and demonstrates good preparation for the planned experiments.
The results presented in the paper are for the most part unquestionable from the point of view of their presentation; the well-prepared graphs and drawings, as well as the illustration with photographs, allow the reasoning to be followed. Only the representation of the level of difficulty is gelatinize, by means of photographs seems to be relatively unreliable - perhaps any method determining the level of turbidity would have worked better.
I also very much appreciated the preparation and description of Figure 5 aimed at showing the relationships in sucrose metabolic pathways involved in response to high temperature.
The well-prepared discussion, referring to other literature data, shows a critical approach to the results obtained.
In my opinion, the article presented is worthy of publication.
Response: Thanks for your professional comments and kind suggestion. Your encouragement will make our research do better. For the gelatinize property of HT-treated starch showed in figure 2F, the HT-treated endosperm starch was also analyzed by a differential scanning calorimeter (DSC) and the result showed that the gelatinization enthalpy (ΔH) of starch under HT were significantly higher than NT. Starch solubility in urea solutions was also tested, revealing that HT-treated starch was more resistant to gelatinization, even in in 6 M to 9 M urea, while NT starch gelatinized was easier (Figure 2G). We have corrected it in the revised MS in line138-141.